# Eosinophils and Neutrophils—Molecular Differences Revealed by Spontaneous Raman, CARS and Fluorescence Microscopy

**DOI:** 10.3390/cells9092041

**Published:** 2020-09-07

**Authors:** Aleksandra Dorosz, Marek Grosicki, Jakub Dybas, Ewelina Matuszyk, Marko Rodewald, Tobias Meyer, Jürgen Popp, Kamilla Malek, Malgorzata Baranska

**Affiliations:** 1Faculty of Chemistry, Jagiellonian University, Gronostajowa 2, 30-387 Krakow, Poland; aleksandra.dorosz@doctoral.uj.edu.pl; 2Jagiellonian Centre for Experimental Therapeutics (JCET), Jagiellonian University, Bobrzynskiego 14, 30-348 Krakow, Poland; marek.grosicki@jcet.eu (M.G.); jakub.dybas@jcet.eu (J.D.); ewelina.matuszyk@uj.edu.pl (E.M.); 3Institute of Physical Chemistry (IPC) and Abbe Center of Photonics (ACP), Friedrich-Schiller-University, Helmholtzweg 4, 07745 Jena, Germany; marko.rodewald@uni-jena.de (M.R.); juergen.popp@uni-jena.de (J.P.); 4Leibniz Institute of Photonic Technology e.V. Member of Leibniz Health Technologies, Jena Albert-Einstein-Str. 9, 07745 Jena, Germany; tobias.meyer@leibniz-ipht.de

**Keywords:** eosinophils, neutrophils, eosinophil peroxidase, myeloperoxidase, lipid bodies, Raman microscopy, coherent anti-Stokes Raman scattering (CARS), fluorescence microscopy

## Abstract

Leukocytes are a part of the immune system that plays an important role in the host’s defense against viral, bacterial, and fungal infections. Among the human leukocytes, two granulocytes, neutrophils (Ne) and eosinophils (EOS) play an important role in the innate immune system. For that purpose, eosinophils and neutrophils contain specific granules containing protoporphyrin-type proteins such as eosinophil peroxidase (EPO) and myeloperoxidase (MPO), respectively, which contribute directly to their anti-infection activity. Since both proteins are structurally and functionally different, they could potentially be a marker of both cells’ types. To prove this hypothesis, UV−Vis absorption spectroscopy and Raman imaging were applied to analyze EPO and MPO and their content in leukocytes isolated from the whole blood. Moreover, leukocytes can contain lipidic structures, called lipid bodies (LBs), which are linked to the regulation of immune responses and are considered to be a marker of cell inflammation. In this work, we showed how to determine the number of LBs in two types of granulocytes, EOS and Ne, using fluorescence and coherent anti-Stokes Raman scattering (CARS) microscopy. Spectroscopic differences of EPO and MPO can be used to identify these cells in blood samples, while the detection of LBs can indicate the cell inflammation process.

## 1. Introduction

Eosinophils (EOS), neutrophils (Ne) and basophils are granulocytes found in human peripheral blood and tissues. They are small and round-shaped with specific granules in their cytoplasm, which contain characteristic proteins specific for each granulocyte type. Granulocytes differ also in the shape of their nuclei and their physiological functions [1]. Eosinophils are round-shaped immune cells with a diameter of 10–20 μm, having a bilobed nucleus and large acidophilic cytoplasmic granules [2,3,4,5]. The abundance of EOS in the peripheral blood is about 1–6% of all white blood cells and their lifespan in blood is about 18 h [4]. Approximately 200 granules with a diameter of 0.9–1.3 μm present in the EOS cytoplasm contain a pull of proteins including major basic protein (MBP), eosinophil derived neurotoxin (EDN), eosinophil cationic protein (ECP), and—eosinophil peroxidase (EPO) [6,7]. Eosinophils are involved in the initiation and propagation of inflammatory responses [8], helminth infections and allergic diseases [7]. They are mainly known for their antiparasitic activity, but they also constitute a prevalent cell population in the female reproductive tract [3]. EOS cells secrete an array of cytokines, which promote T cell proliferation, activation and polarization [3,7] and also participate in the communication between other cells of the immune system. The number of eosinophils increases in response to allergies and parasitic infections [6]. Neutrophils are in turn azurophilic granulocytes with a diameter of 10–12 μm and multilobed nuclei [9]. They are the most abundant leukocytes (54–62%) and live in circulation for about 6–12 h. Neutrophils migrate to infection sites caused by bacteria [10] and other microorganisms, where they participate in their phagocytosis [9,11]. Myeloperoxidase (MPO) is accumulated in the matrix of granules surrounded by trilaminar membrane [2,12,13].

Both the most characteristic proteins, EPO and MPO, are protoporphyrins and belong to the group of mammalian peroxidases. EPO consists of two subunits, a heavy chain (50–58 kD) and a light one (10.5–15.5 kD) in 1:1 stoichiometry [2]. The prosthetic group of EPO is a high spin, hexa-coordinated, ferric protoporphyrin IX. Major functions of EPO include the catalysis of oxidation of halides, pseudohalidase and nitric oxide to highly reactive oxygen species with cytotoxic properties [6]. MPO is also built of two monomers, which are linked through cysteine bridges [14]. One heavy subunit is covalently bonded with the heme ring, while glycosylated light chains contain a modified iron protoporphyrin IX active site [14,15]. MPO in combination with H_2_O_2_ and halides is involved in killing viruses, and bacteria. EPO also shows antibacterial activity but only in the presence of bromides [2,15,16,17]. In order to differentiate EOS and Ne cells, their peroxidases have been analyzed by employing Raman spectroscopy, in particular resonance Raman spectroscopy. Spectra measured under resonance Raman conditions require the use of laser excitation in the range of electronic absorption of a chromophore. A few reports have shown this approach by illuminating EOS and Ne with laser lines at 406, 514, and 660 nm [11,18,19]. However, these previous studies lack details about the molecular structure of EPO alone [8,18] or a certain group of compounds found in EOS and Ne, e.g., lipids [2]. In our work we employed spontaneous Raman spectroscopy with three different excitation wavelengths (532, 633, and 785 nm) in broad spectral ranges that allowed us a detailed evaluation of the molecular properties of EPO and MPO, both on-resonance (532 and 633 nm) and pre-resonance (785 nm), as well as to deliver information about DNA, protein and lipid composition of EOS and Ne cells.

Lipids constituting lipid bodies (LBs) in granulocytes can be recognized relatively easily by Raman spectroscopy due to the large Raman cross-section of long nonpolar acyl chains (highly polarizable C–H and C–C bonds) [20,21]. The occurrence of LBs in leukocytes is associated with infection and inflammatory responses during which the increased size and number of LBs are considered as a marker of the leukocyte activation [22,23]. In principle, LBs are rapidly formed and secrete arachidonyl phopholipids used next in the production of eicosanoids, which are inflammatory mediators [24,25,26].

Raman spectroscopic imaging is a powerful tool in the analysis of LB composition but a high through-put requires scanning of a large number of cells since LBs occur in a few percent of granulocytes only [6]. Therefore, the use of other methods, like fluorescence and coherent anti-Stokes Raman scattering (CARS) microscopies, is desired to visualize the distribution of LBs in granulocytes. The identification of LBs by fluorescence microscopy requires staining of the cells with dyes specifically accumulated in lipidic cellular compartments, e.g., Nile Red or BODIPY [27], whereas CARS relies on multiphoton, nonlinear processes and provides sensitive, rapid and label-free detection of lipids even in living cells [28,29,30,31,32,33,34]. Moreover, the sensitivity of CARS is better than that of spontaneous Raman scattering [33], therefore, we decided to apply CARS microscopy to understand the metabolism and regulation of lipid storage in cells and tissues in the course of diseases [28,30,31].

Here, we provide a comprehensive chemical characterization of eosinophils and neutrophils by means of spectroscopic imaging techniques to give a panel of markers suitable for further differentiation of these granulocytes and to follow stress conditions. Firstly we examine EPO and MPO by spontaneous Raman spectroscopy with laser excitations at 532, 633, and 785 nm in order to fully understand the complexity and differences between these two proteins which could be useful for differentiation between EOS and Ne. Electronic absorption spectra were collected to define precisely the conditions of resonance and pre-resonance Raman scattering for reference peroxidases and granulocytes. In the next step of analysis, Raman images of eosinophils and neutrophils were measured with the same three laser excitations. Beside protoporphyrins, also LBs were analyzed, with regard to their number and chemical composition. The former were studied by fluorescence and CARS microscopy.

## 2. Materials and Methods

### 2.1. Chemicals

EPO solution (LeeBiosolutions, Maryland Heights, MO, USA), MPO solution (LeeBiosolutions, Maryland Heights, MO, USA), EPO powder (Sigma-Aldrich, St. Louis, MO, USA), MPO powder (LeeBiosolution, Maryland Heights, MO, USA), dextran from Leuconostoc spp. 500,000 (Sigma-Aldrich, St. Louis, MO, USA), Ficoll-Paque Plus d = 1.077 g/mL (GE Healthcare, Chicago, IL, USA), human eosinophil isolation kit (MiltenyiBiotec, Bergisch Gladbach, Germany), DPBS (Dulbecco’s phosphate-buffered saline, Gibco, ThermoFisher, Waltham, MA, USA), 10% fetal bovine serum (FBS, Gibco, ThermoFisher, Waltham, MA, USA), penicillin-streptomycin (10,000 U/mL, ThermoFisher Scientific, Waltham, MA, USA), Roswell Park Memorial Institute (RPMI) 1640 medium with L-glutamine and 25 mmol/l HEPES (Gibco), EDTA (Sigma-Aldrich, St. Louis, MO, USA).

Reference eosinophil peroxidase (EPO, Munich, Germany) and myeloperoxidase (MPO) were purchased as lyophilized powders and solutions (LeeBiosolutions, Maryland Heights, MO, USA). EPO solution (≥98% purity) was dissolved in 0.1 M sodium phosphate solution with 0.4 M sodium chloride (pH 7.5). MPO solution (≥96% purity) was dissolved in 50 mM sodium acetate and 100 mM sodium chloride solution with preservative (bromo-nitro-dioxane, pH 6.0).

### 2.2. Isolation of Eosinophils and Neutrophils from Blood

Human eosinophils and neutrophils were isolated according to the method described by Grosicki et al. [35]. Briefly, human peripheral blood, from which cells were isolated, was purchased from the Regional Blood Transfusion Center (Krakow, Poland). Blood was collected in heparin-coated tubes. Erythrocytes (RBCs) were discarded from the remaining blood cells through 1% dextran solution sedimentation. Peripheral blood mononuclear cells (PBMCs) were removed by Ficoll−Paque density gradient separation. Remaining RBCs were removed through hypotonic shock lysis. Eosinophils and neutrophils were further purified from the remaining cells using an immunomagnetic cell separation method (human eosinophil isolation kit, Easy Step), according to the manufacturer’s protocol. The purity of isolated eosinophils was estimated based on flow-cytometric methodology at 97%, remaining cells consisted of neutrophils. According to Polish law, blood samples purchased from the Regional Blood Transfusion Center do not require the permission of an Ethics Commission.

### 2.3. Spectroscopic Measurements

#### 2.3.1. UV−Vis Absorption Spectroscopy

Reference solutions of EPO and MPO (concentrations ca. 0.5–1.5 and 1.0–2.0 mg/mL, respectively) as well as eosinophils and neutrophils (ca. 300,000 cells suspended in 1 mL PBS) were measured by using a Lambda 950 spectrophotometer (PerkinElmer, Waltham, MA, USA). Electronic spectra were acquired in the 200–800 nm range with a step size of 1 nm and accumulation of 1 scan in a 1 cm quartz cuvette. Spectra were baseline corrected using a rubber band method with 64 baseline points (OPUS 7.2 software).

#### 2.3.2. Raman Spectroscopy

Raman spectra of reference EPO and MPO and isolated cells were recorded using a WITec confocal CRM alpha 300 Raman microscope equipped with air-cooled lasers operating at 532, 633, and 785 nm. The CCD detector was cooled to −60 °C. The lasers were coupled to the microscope via a single mode optical fiber with a diameter of 50 μm (for 532 nm) and 100 μm (for 633 nm and 785 nm). Reference samples were illuminated through an air Olympus MPLAN objective (100 × /0.9 NA) while the cells were measured with a water immersion Nikon Fluor objective (60 × /1.0 NA). All samples were placed on CaF_2_ windows.

In order to avoid photothermal degradation of peroxidases, the laser power was adjusted individually for each reference sample, i.e., for the 532 nm laser line: 167 and 11.37 μW, for the 633 nm laser line: 0.5 and 10 mW, and for the 785 nm laser line: 10 and 40 mW for MPO and EPO, respectively. Adjustment of the laser power was performed applying a live scan measurement mode with the aim to obtain the best signal-to-noise ratio without affecting the susceptible peroxidases’ structure and was set on the highest possible value which provided unaltered Raman spectrum. The spectra were recorded at 10 randomly chosen positions with a 30 s integration time and 10 accumulations for all excitation wavelengths. Spectra were preprocessed with cosmic spike removal (filter size 3 and dynamic factor 8), vector normalized and then averaged, using the WITec Project Plus software.

Measurements of cells were performed using the same three laser lines (532, 633 and 785 nm) and a power at the sample site of 30, 20 and 130 mW, respectively. Raman spectra were acquired in randomly chosen single points and as Raman images. For the former, 5 spectra per cell (15 cells in total) were recorded with 1 s integration time and 10 accumulations. Raman imaging of 10 cells was performed with a 0.5 μm step size, giving a spatial resolution of ca. 1–1.5 μm, and with 1 s integration time for a single spectrum. *K*-means cluster analysis was performed after cosmic spike removal (filter size 3 and dynamic factor 8) and baseline correction (polynomial degree of 3) for each imaged cell using the Manhattan-distance formulation to separate nuclei and cytoplasm region. The analysis was performed with the WITec Project Plus software. The Raman bands position was obtained by reading the positions of the maxima using the WITec software.

#### 2.3.3. Fluorescence Microscopy

Fluorescence images were obtained by using an Olympus Scan^R automated fluorescence microscope. Cellular nuclei were stained with Hoechst 33,342 (ThermoFisher, Waltham, MA, USA) (360/497 nm), whereas LBs were stained with BODIPY 493/503 (ThermoFisher, Waltham, MA, USA) (493/504 nm). Briefly, cells were placed in a poly-L-lysis solution and centrifuged (300 g, 5 min at room temperature) to attach cells to the bottom of the 96-well plate. Cells were fixed with 4% paraformaldehyde for 10 min. Afterwards, cells were washed gently with DPBS (Gibco) and stained with 4 µg/mL BODIPY 493/503 dye for 30 min at room temperature. Next, washed cells were stained with Hoechst 33,342 (0.5:1000) and imaged using the fluorescence microscope. Fluorescence images were analyzed using a Columbus 2.4.2 image data storage and analysis system software (PerkinElmer, Waltham, MA, USA). Data were collected from 4 independent biological experiments.

#### 2.3.4. Coherent Anti-Stokes Raman Scattering Microscopy (CARS)

The experimental platform for CARS imaging has been described in detail previously [36]. In a nutshell, a frequency doubled continuous wave Nd-Vanadate laser (Verdi-V18 Coherent, Santa Clara, CA, USA) at 532 nm was used to pump a Ti:sapphire laser (Mira HP Coherent, Santa Clara, CA, USA) to generate ps pulses at 76 MHz pulse repetition rate and a wavelength of 832(2) nm. A 45:55 R:T beam splitter was employed to divide the laser into two separate beams. The reflected part of the beam was used as the Stokes beam, the transmitted part pumped an optical parametric oscillator (OPO, APE, Berlin, Germany) in order to generate the tunable pump beam by parametric conversion into the near-infrared region (NIR) and subsequent frequency doubling by second harmonic generation (SHG). The SHG of the signal emission can be tuned from 500 to 700 nm. In order to address the symmetric CH_2_ stretching vibration at 2850 cm^−1^, the OPO beam was tuned to 672.5(7) nm. To recombine Stokes and pump beams and optimize their spatial and temporal overlap, a delay stage was applied in combination with a retro reflector and an 800 nm long pass dichroic mirror. The beams were directed into the laser scanning microscope (LSM 510 Meta, Zeiss, Oberkochen, Germany). The laser radiation was later focused on the samples through a 40x (1.1 NA) water immersion objective (LD-C-Apochromat, Zeiss, Oberkochen, Germany). The average laser powers at the sample site were 48 mW (pump beam) and 39 mW (Stokes beam). In this experiment the CARS signal was collected in a forward direction by an NA 0.8 condenser and separated from the residual pump and Stokes light by dielectric filters (band pass 550/88, short pass 650 nm, Semrock, New York, NY, USA) and was detected by a photomultiplier (Hamamatsu R6257, Kamioka, Japan). CARS images were recorded in a field of view of 112.5 × 112.5 µm^2^ with a resolution of 1024 × 1024 pixels at an integration time of 1.60 µs per pixel. Sixteen frames were averaged for each image.

## 3. Results

### 3.1. Resonance Raman Spectra of EPO and MPO in Solutions

The electronic absorption spectra of the two proteins exhibited some differences (Figure 1B). Among electronic transitions of EPO, a strong Soret band was observed at 412 nm while Q_v_, Q_0_ and Q_I_ as well as CT (charge-transfer) absorbances appeared at 507, 546, 591, and 640 nm, respectively [11,19,37,38]. A band at 741 nm was probably an artefact from the aperture or a lamp. In the spectrum of MPO these bands were shifted, i.e., the Soret band was present at 428 nm, Q_v_ and Q_0_ absorbances were found at 499 and 572 nm, respectively, while the CT absorbance was located at 623 nm. An additional band at 687 nm was of unknown origin [16,38,39], while the band at 740 nm was again an artefact. The spectral variations can be attributed to the slightly different molecular structures of the porphyrins and surrounding polypeptide chains of these proteins (Figure 1A) [11,19,37]. Figure 1C shows the comparison of Raman spectra of the reference EPO and MPO solutions recorded with excitations at 532, 633 nm and 785 nm. Since 785 nm Raman spectra of EPO and MPO solutions showed the presence of solvent bands, we included here the spectra of the solid peroxidases (black line).

Resonance Raman spectra of EPO and MPO solutions with an excitation in the region of charge-transfer electronic transition (633 nm, Figure 1C) allowed the differentiation of these peroxidases by a set of bands specific for EPO, i.e., at 937 (ν_46_), 1122 (ν_22_), 1216 (ν_13_) and 1610 cm^−1^ ν(C=C) as well as for MPO, i.e., at 933 (ν_46_), 1115 (ν_22_), 1240 (ν_42_) and 1534 cm^−1^ (ν_38_). Some bands appeared in the spectra of both proteins, i.e., at 680 (ν_7_), ca. 750 (ν_15_), 1005 (phenylalanine), and 1177 cm^−1^ (ν_30_) [11,19,40]. In the Raman spectrum of EPO, one can observe a triplet of signals at 1314 (ν_21_), 1341 (δ_s_(=CH_2_) out of phase), and 1395 cm^−1^ (ν_40_), in which the middle band was the most intense. In turn, a quartet of bands at 1314 (ν_21_), 1341 (ν_41_), 1361 (ν_4_), and 1395 cm^−1^ (ν_40_) was present in this range of the MPO spectrum. The intensities of these bands differed only slightly [11,19,40].

In the 532 nm spectra of EPO new bands could be observed, which were not visible or hardly visible, when measured with 633 nm excitation, e.g., 1266 (CH_2_ wagging) and 1610 cm^−1^ ν(C=C), while for MPO a band at 1395 cm^−1^ (ν_40_) could be noticed. The use of different excitations allowed the enhancement of different vibrations in the molecule which, in the case of EPO, could be observed for the bands at 1122 (ν_22_), 1361 (ν_4_), and 1555 cm^−1^ (ν_11_), in the case of MPO, for the bands at 1336 (ν_41_), 1367 (ν_4_), 1395 (ν_40_), 1555 (ν_11_), 1588 (ν_2_), and 1610 cm^−1^ [ν(C = C], c.f. Table 1 [11,19,40]. Some of these bands appeared as weak shoulders or single bands of low intensity.

### 3.2. Electronic and Raman Features of Isolated Eosinophils and Neutrophils

Human EOS and Ne cells were isolated from the blood of healthy donors. Cell suspensions in phosphate-buffered saline were examined by electronic and Raman spectroscopy, similarly to reference EPO and MPO standards (Figure 2). In our investigations, a broad Soret band of neutrophils was located at 448 nm (data not shown). A Q_0_ band of cyt b_558_ was observed in turn at 565 nm (Figure 2A) [43]. Other weak bands found in the UV−Vis spectrum of Ne most likely originated from other hemoproteins occurring in neutrophils. Considering laser excitations at 532 and 633 nm, we expected that Raman spectra of EOS and Ne would show resonance Raman spectroscopy (rR) enhancement for EPO and MPO, originating from the Q_v_ and CT electronic transitions, respectively.

The Raman spectra of isolated cells, collected with excitations at 532, 633 and 785 nm, are displayed in Figure 2B. The Raman spectra of EOS and Ne exhibited the presence of Raman bands typically assigned to cellular biomacromolecules, i.e., at 977 (CH_3_ deformation of lipids, proteins), 1005 (phenylalanine), 1240–1254 (amide III), 1309 (δ(CH_2_) of lipids, amide III), 1330–1344 (ring breathing modes of adenine, guanine, δ(C-H)), ca. 1335/1445 (δ(CH_2_/CH_3_) of proteins and phospholipids), and 1659 cm^−1^ (amide I: α-helical conformations, ν(C = C): lipids). The most prominent bands of hemoproteins appeared as shoulders of the amide I band in the region of 1500–1610 cm^−1^ and as a band at ca. 756 cm^−1^, regardless of the wavelength of the laser line. Other bands labelled in green and blue in the spectra of EOS and Ne, respectively, showed a contribution of modes assigned to EPO and MPO bands as well as to cellular biomolecules, see Table 1, Figure 1C and Figure 2B.

Even though the 785 nm excitation line was not close enough to electronic transitions of neutrophils and eosinophils cells for rR, we found low-intensity bands attributed to their peroxidases; i.e., at 756 (ν_15_), 1122 (ν_22_) and 1580 cm^−1^ (ν_2_) for EPO and at 756 (ν_15_), 933 (ν_46_), and 1588 cm^−1^ (ν_2_) for MPO. However, their intensities hindered a clear discrimination of both types of granulocytes_._

Laser illumination of granulocytes in the CT electronic transition (633 nm) resulted in the enhancement of EPO and MPO bands comparing to the nonresonant conditions at 785 nm (Figure 2B). Clearly, the most characteristic bands, which can be used to distinguish the cells, appeared at 1117 (ν_22_), 1216 (ν_13_), and 1564 cm^−1^ (ν_19_) in the spectrum of EOS and at 1115 (ν_22_), 1240 (ν_42_), 1534(ν_38_), and 1580 cm^−1^ (ν_2_) for Ne. The intensities of the porphyrin bands are higher for eosinophils than for neutrophils due to a higher concentration of EPO than MPO in the respective cells.

The resonance Raman effect was also observed in the spectra of the granulocytes after excitation at 532 nm, i.e., in the region of the Q bands of EPO and MPO (Figure 1B and Figure 2A). The presence of the two proteins was confirmed by weak-medium bands located at 756 (ν_15_), 1177 (ν_30_, EPO only), 1555 (ν_11_), 1588 (ν_2_), and 1610 cm^−1^ ν(C = C, EPO only). Other bands of peroxidases were overlapped by bands of cellular components. Similarly to EPO and MPO in solution, the hemoproteins in the cells possessed a 6-coordinated high spin ferric heme group [11,19,40].

In addition, we performed Raman imaging of 15 eosinophil and neutrophil cells with excitation lines at 532 and 633 nm. Figure 3 illustrates results of uni- and multivariate analysis (*k*-means cluster analysis) performed for Raman images of the cells. Raman distribution images were constructed by integration of the high-wavenumber region (2800–3030 cm^−1^) showing the presence of the stretching modes of the CH/CH_2_/CH_3_ groups of all organic components in the cells and by integration of the 780–800 cm^−1^ region indicative of the nucleus (breathing vibrations of thymine and cytosine with the O-P-O backbone mode) (Figure 3A) [44].

KMC analysis was performed in order to segregate pixel Raman spectra into two main subcellular compartments, i.e., cytoplasm and nuclei (Figure 3B). Nuclei were identified by the DNA marker bands at 790 (breathing vibrations of thymine and cytosine with O-P-O backbone mode), 1093 (symmetric stretching vibration of the PO_2_ group), 1261/1311/1344 (CH deformations of T, A and G), and 1483/1582 cm^−1^ (stretches of the G and A rings) [18,45,46].

### 3.3. Fluorescence and Coherent Anti-Stokes Raman Scattering Imaging of Human Eosinophils and Neutrophils

Staining of cellular structures combined with fluorescence microscopy is the classical tool to illustrate morphological properties of cells such as the presence and the shape of the nuclei (a Hoechst dye) as well as the content of LBs (a BODIPY dye), see Figure 4A, upper panel. To examine these images in detail, a Columbus image data storage and analysis system was employed. It enabled a multi-parametric analysis of fluorescence intensity recorded for a hundred cells and provided a set of geometrical parameters of the stained cellular structures such as spot area, symmetry, radius, etc. (Figure 4B). Obtained images exhibited the presence of both cellular structures in EOS and Ne cells, but the 20× lens magnification of the CQ1 confocal quantitative image cytometer did not reveal a clear-cut visualization of the shape and number of the selected cell constituents. Based on the fluorescent images and BODIPY staining (aimed at LB detection), highly fluorescent and aggregated BODIPY signals were detected on human eosinophils. On the other hand, such signals were not detected on human neutrophils, where BODIPY staining was dispersed throughout the neutrophil cytoplasm. As such, the fluorescent imaging with accessible magnification was not sufficient for correct LB estimation. For that the CARS imaging was implemented.

To show the distribution of lipids in the granulocytes, we employed the BODIPY staining for fluorescence microscopy and label-free CARS imaging detecting the 2853 cm^−1^ band of the symmetric stretching mode of the methylene groups in the acyl fatty acids chains (Figure 4).

## 4. Discussion

Since EPO and MPO are hemoproteins containing a chromophore group, depending on the laser excitation used (532, 633 and 785 nm) for Raman measurements it is possible to be in resonance or preresonance conditions [11,19,40] (Figure 1C). In the resonance Raman (rR) spectra we observed enhancement of Raman intensities of certain modes when an excitation wavelength was selected from the region of electronic transition. According to the resonance mechanism it can be distinguished between A-, B- and C-type enhancement depending on the excitation wavelength used and the type of electronic transition. Excitation in the Soret band region (ca. 400 nm) resulted in enhancement of totally symmetric modes (type A) while excitation wavelengths closer to the Q bands (500–600 nm, here at 532 nm) yielded enhancement of both, totally and nontotally symmetric modes with predominance of the latter (type B). C-type occurred upon excitation with energies matching the vibronic side band of a forbidden or weakly allowed electronic transition, i.e., Q_v_ and Q_IV_ regions and led to the enhancement of overtones and combination modes [47,48]. Excitation in the CT transition region (ca. 620 nm, here at 633 nm) resulted in the enhancement of modes related to the porphyrin ring and iron-ligand vibrations [49,50,51].

In Figure 2A,B we matched the excitation laser lines to the energy close to Q_0_ and CT electronic transitions, i.e., 532 and 633 nm, respectively, as well as to non-resonant conditions (785 nm). This part of our experiment was focused on the determination of the Raman features of the proteins, which were used for comparison with the in situ Raman signature of the hemoproteins in granulocytes.

A visual comparison of EPO and MPO rR spectra collected with excitations at 532 and 633 nm showed that the CT-associated Raman spectra clearly differentiated both proteins and these differences were mainly related to the bands’ positions and their relative intensities (Figure 1, Table 1). On the other hand, the 532 nm spectra of both proteins were very similar and thus it was difficult to distinguish EPO and MPO, especially when detected inside cells where such minor changes can be easily missed. However, 633 and 785 nm excitations provided significantly different Raman spectra with a specific pattern for each protein and therefore both excitation wavelengths could be employed for unambiguous recognition of EPO and MPO and further for differentiation of granulocytes. Since both proteins are mammalian peroxidases and possess heme groups, their Raman spectra exhibit similar profiles. However, differences in peripheral substituents of their prosthetic groups can be recognized when rR and specific laser excitation are applied.

In addition, resonance Raman spectra of EPO and MPO acquired with excitation wavelengths in the visible region of the light indicated coordination number, spin, and oxidation states of the iron in the porphyrin unit. Bands at 1367 (ν_4_), 1555 (ν_11_) and 1610 cm^−1^ ν(C = C) suggest that both peroxidases possess a hexa-coordinated, high spin ferric heme group in solution [11,19,40].

Q_v_, Q_0_ and CT absorption maxima in EOS and Ne cells appeared at similar positions as in the spectra of EPO and MPO, only the intensity of the Q_v_ band (ca. 500 nm) increased compared to reference peroxidases, cf. Figure 1 and Figure 2. No bands appeared above 700 nm. Neutrophilic granulocytes contain two heme proteins, MPO as well as cytochrome b_558_, but they can be differentiated only by the position of the Soret bands located at 430 and 413 nm, respectively [14]. In our study, the electronic spectrum of EOS cells showed the presence of an additional band at 687 nm that was absent in the spectrum of neat EPO, see Figure 1 and Figure 2. Except EPO, eosinophilic granules contain major basic protein (MBP), ribonucleases eosinophil cationic protein (ECP) and eosinophil-derived neurotoxin (EDN), but none of these proteins contain the heme moiety [4]. There is no report showing their electronic absorption characteristics, so we cannot exclude their contribution to these bands.

Interestingly, bands of EPO and MPO in the Raman spectra of isolated cells were not dominant in any of the spectra of the granulocytes as observed previously by Salmaso [19] and Sijtsema [41] and their coworkers (Figure 2B). The nucleus classes in the KMC maps clearly differentiated bi- and multilobed nuclei typical for eosinophils and neutrophils, respectively (Figure 3B). Their spectral characteristics were identical, highlighting the lack of chemical differences between their nuclei. In turn, the classes assigned to cytoplasm exhibited the presence of typical cellular components as well as EPO and MPO whose bands’ positions were similar to the single point Raman spectra of the cells, cf. Figure 2 and Figure 3. As previously, the 532 nm resonance Raman spectra of EOS and Ne cytoplasm were virtually identical whereas excitation at 633 nm indicated that the hemoproteins in the cells could be recognized in the 1500–1620 cm^−1^ region. We noted that pixel Raman spectra collected in imaging mode exhibited a lower intensity of rR bands of both hemoproteins compared to single point measurements as a result of the 10-times shorter accumulation time per spectrum for imaging measurements. However, the signal-to-noise ratio was better in the imaging mode, due to performing KMC analysis and averaging spectra into classes. Taking also into account the total time required for the collection of Raman images, we propose that the clinical differentiation of the granulocytes should preferably be performed by using single point measurements from cells.

As discussed in the introduction, eosinophils and neutrophils are mostly differentiated based on the nucleus shape, granule content as well as the presence of lipid bodies (LBs) observed in optical and fluorescence microscopy as we found in Raman images depicted in Figure 3 [6,13,52]. In turn the fluorescence images of EOS and Ne stained with the Hoechst dye and combined with the Columbus analysis showed that these organelles mainly differed in their symmetry whereas the radius and compactness were almost identical. Parameters calculated for the nucleus symmetry indicated that both types of nuclei were not oval and divided into two and three segments, in EOS and Ne cells, respectively.

We have previously reported the sensing capability of normal Raman scattering imaging for the detection of LBs in primary eosinophils and eosinophilic cell line EoL-1 [6]. Nevertheless, LBs are not formed in each cell under normal conditions and without additional stimulation, and thus the collection of Raman images of single cells to detect LBs is time consuming. The CARS recognized LBs in some EOS cells (brighter spots in the CARS image, Figure 4A) in contrast to Ne. For the latter the CARS image visualized the shape of cells with granules evenly distributed near the cell surface. In turn, fluorescence microscopy detected lipids in the EOS as well as Ne cells upon staining with BODIPY, but their distribution was not clearly resolved and was overlapped by the Hoechst signal in the nuclei. On the other hand, the multi-parametric analysis of the BODIPY fluorescent intensity with the use of the Columbus software indicated that this signal is higher in EOS than in Ne (Figure 4C). A small spot area in the EOS cells was assigned to LBs and this parameter, as well as the intensity ratio of the Hoechst and BODIPY dyes, was used to calculate an average number of LBs in all detected cells amounting to two LBs per cell (Figure 4B). In the case of neutrophils, lipids were rarely accumulated as droplets. Both fluorescence and CARS images revealed a similar conclusion about the distribution of lipids and LBs in the cytoplasm of examined cells.

## 5. Conclusions

Raman (spontaneous, resonance and coherent anti-Stokes scattering) and fluorescence microscopies were used together for the first time for the recognition of biochemical features of eosinophils and neutrophils—the main players in fighting infections and inflammation of the body. The examination of spectroscopic properties of eosinophil peroxidase and myeloperoxidase present in granules of the cells indicated that resonance Raman conditions are required to differentiate the EOS and Ne cells. Both peroxidases can be detected upon illumination with VIS and NIR lasers commonly used in Raman microscopy, regardless of the mode of data collection, i.e., single point spectra or cell Raman images. All of them determined in situ that iron ion in the heme group is hexa-coordinated with a high spin configuration. However, specific spectral markers of EPO and MPO appeared only upon excitation of charge-transfer electronic transitions at 633 nm. In turn, Raman microscopy provided comprehensive information about the chemical composition of cytoplasm and nucleus, including phospholipids, proteins and DNA constituents. This label-free analysis was complemented by target-oriented CARS and fluorescence microscopy to evaluate the distribution of lipids in the granulocytes. Both techniques revealed their presence but a clear-cut identification of lipid bodies in eosinophils was observed in the CARS images only. These lipidic reservoirs were unevenly distributed among EOS cells and multi-parametric morphological analysis of fluorescence staining estimated the average number of LBs per cell to be two. In neutrophils, the lipids were spread out in the entire cytosol. We propose here that the clinical differentiation of the granulocytes should be preferably performed using single point measurements from cells, which give similar results as imaging but are considerably faster. The obtained preliminary results may lead to further investigations into the identified biomarkers for classification of the granulocytes and for a better understanding of the physiology behind them under stress conditions and diseases.

## Figures and Tables

**Figure 1 cells-09-02041-f001:**
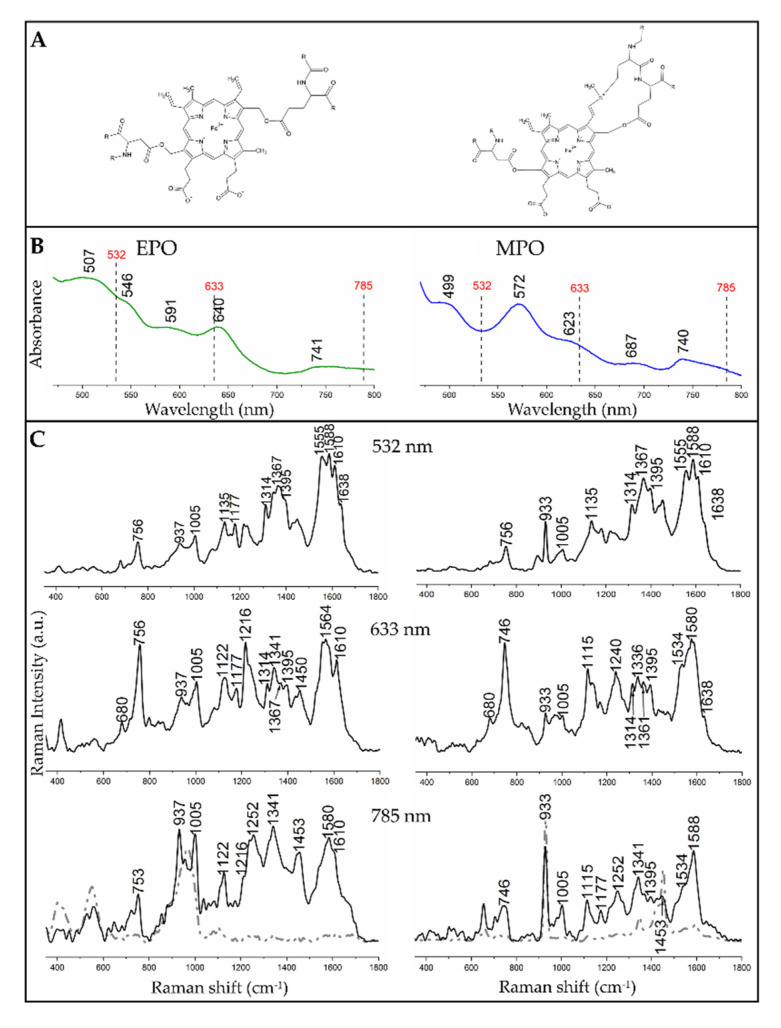
Spectral characteristics of neat eosinophil peroxidase (EPO, left) and myeloperoxidase (MPO, right) with their molecular structures. (**A**) Structures of the proteins. (**B**) UV−Vis absorption spectra of EPO and MPO (strong Soret bands at ca. 410–420 nm were not displayed to improve visibility of absorption bands in the region of the used excitation lines). (**C**) Raman spectra of EPO and MPO solutions collected with different excitation wavelengths. For the 785 nm laser excitation, the black trace represents the spectra of the solids, while the grey dashed lines are the spectra of the solutions.

**Figure 2 cells-09-02041-f002:**
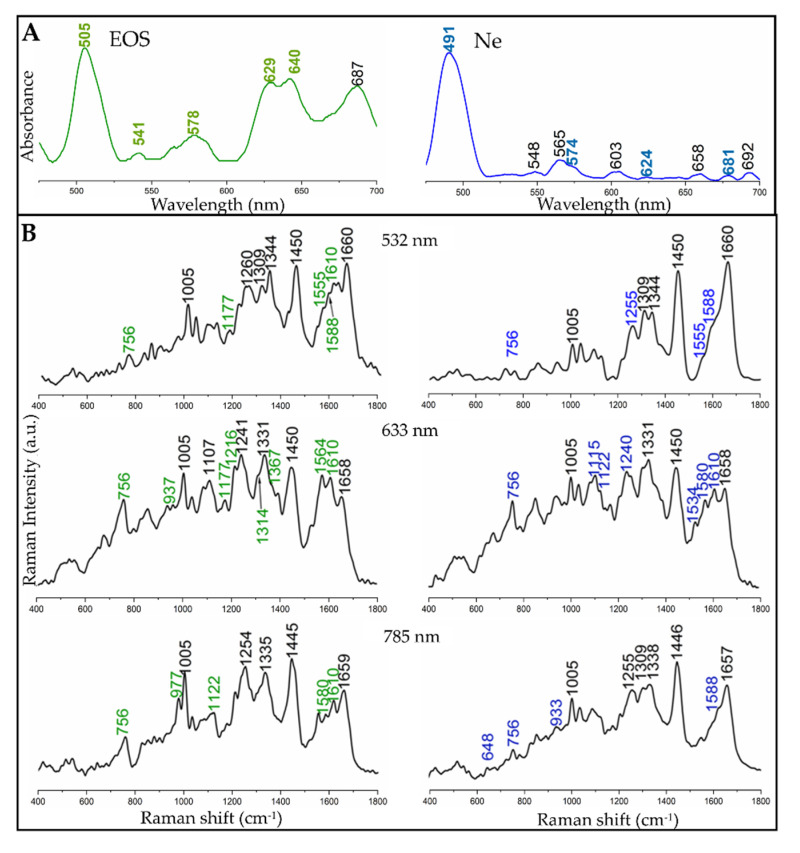
Spectral characteristics of eosinophil (EOS, left) and neutrophil (Ne, right) cells. (**A**) UV−Vis electronic spectra of suspended EOS and Ne cells (strong Soret bands at ca. 410–440 nm are not shown to improve visibility of absorption bands in the region of the used excitation lines). (**B**) Averaged single point Raman spectra of EOS and Ne cells (*n* = 10), acquired with excitation wavelengths at 532, 633, and 785 nm. Bands labelled in green and blue correspond to the bands observed in the spectra of EPO and MPO, respectively, (Figure 1). In grey there is presented the spectra of PBS.

**Figure 3 cells-09-02041-f003:**
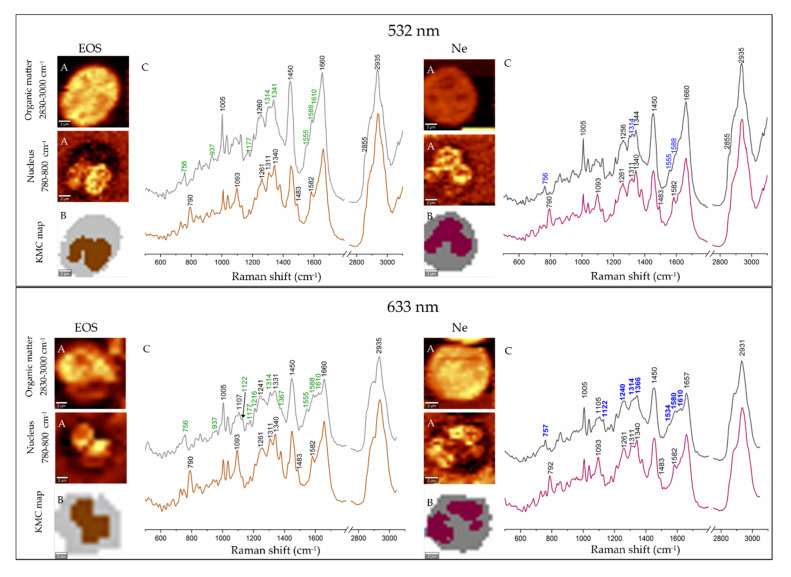
Raman imaging of eosinophil (EOS, left panel) and neutrophil (Ne, right panel) with laser excitations at 532 and 633 nm. (**A**) Raman distribution images of organic matter and nuclei. (**B**) False-color k-means cluster analysis (KMC) maps with the distribution of cytoplasm (grey) and nuclei (brown and purple); (**C**) Mean Raman spectra extracted from KMC analysis (colors of spectra correspond to the colors of KMC classes). Bands labelled in green and blue are assigned to EPO and MPO, respectively. Scale bar: 2 μm.

**Figure 4 cells-09-02041-f004:**
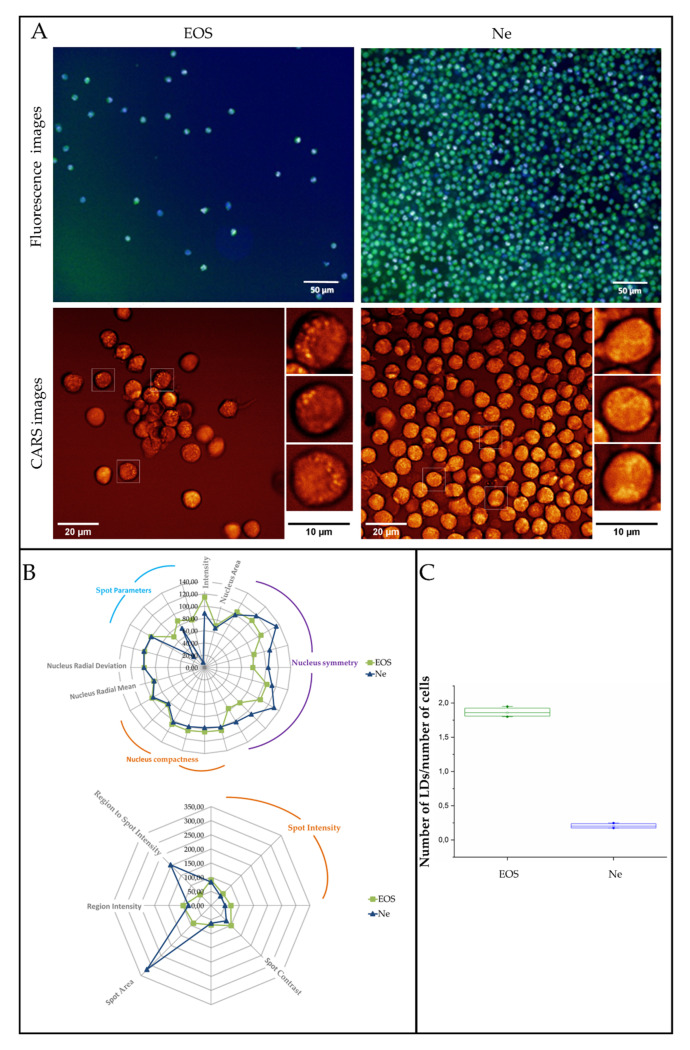
(**A**) Fluorescence (Hoechst 33,342—blue, nuclei and BODIPY 493/503—green, LBs) and coherent anti-Stokes Raman scattering images (CARS) of human eosinophils and neutrophils. (**B**) Multi-parametric morphological analysis of nuclei (upper) and LBs (lower) in fluorescence images. (**C**) A graph showing the number of LBs per cell, calculated from fluorescence images. Values are given as mean ± standard deviation and are shown in box plots: mean (horizontal line), SE (box), minimal and maximal values (whiskers).

**Table 1 cells-09-02041-t001:** An assignment and symmetry of Raman bands of EPO and MPO observed due to excitations at 532, 633, and 785 * nm [11,19,40,41,42].

532 nm	633 nm	785 nm	Assignment of Modes with Their Symmetry
EPO/MPO	EPO/MPO	EPO/MPO	
756/756	756/746	756/746	ν_15_ (B_1g_)
937/933	937/933	937/933	ν_46_, (E_u_)
1005/1005	1005/1005	1005/1005	Phenylalanine
-/-	1122/1115	1122/1115	ν_22_ (A_2g_)
1135/1135	-/-	-/-	ν_14_ (B_1g_)
1177/-	1177/-	-/1177	ν_30_ (B_2g_)
-/-	1216/-	-/-	ν_13_ (B_1g_)
-/-	-/1240	1252/1252	ν_42_ (E_u_)
1314/1314	1314/1314	-/-	ν_21_ (A_2g_)
-/-	1341/1336	1341/1341-/-	ν_41_ (E_u_)/δ_s_(=CH_2_)
1367/1367	1367/1361	-/-	ν_4_ (A_1g_)
1395/1395	1395/1395	-/1395	ν_40_ (E_u_)
-/-	-/-	1453/1453	CH_2_/CH_3_, δ_s_(=CH_2_)
-/-	-/1534	-/1534	unknown
1555/1555	-/-	-/-	ν_11_ (B_1g_)
-/-	1564/-	-/-	ν_19_ (A_2g_)
1588/1588	-/1580	1580/1588	ν_2_ (A_1g_)
1610/1610	1610/-	1610/-	ν(C = C)
1638/1638	-/1638	-/-	ν_10_ (B_1g_)

Modes of EPO and MPO correspond to the D_4h_ symmetry of the porphyrin ring; δ_s_: symmetric bending vibration; ν(C = C) stretching mode of the vinyl substituent, * the band assignment for 785 nm excitation wavelength with regards to spectra of solid EPO and MPO.

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
