# Peer review of "Eosinophils and Neutrophils—Molecular Differences Revealed by Spontaneous Raman, CARS and Fluorescence Microscopy"

_cells, 2020, doi:10.3390/cells9092041_

Round 1
Reviewer 1 Report
In this manuscript, Dorosz and colleagues apply interesting imaging techniques - Raman spectroscopy and CARS - to compare human eosinophils and neutrophils. Although the subject is of interest, there are critical points in the manuscript that require substantial changes.
Major points
1- In the introduction section, there are several points that need to be clarified:
-Why the authors are studying lipid bodies in eosinophils and neutrophils? The number and distribution of these organelles in these cells are well documented in the literature and Raman and CARS have already been applied to the study of LBs.
-Why EPO was considered the most important eos protein?
-What is the purpose of the following sentence: “Basophils are the least numerous population of blood circulating cells and were not included in this study”? The paper is focused on the study of eosinophils and neutrophils. The sentence about basophils is unnecessary and should be deleted.
-In the sentence: “Granulocytes differ also in the shape of their nuclei… “ In fact, all granulocytes have a lobed nucleus, including basophils. Figure 1 does not add any information to the paper and therefore should be deleted.
-Some general statements need revision or are not adequately supported by the references cited. “Eosinophils are round-shaped immune cells with a diameter of 10-20 μm having a bi-lobed nucleus and large acidophilic cytoplasmic granules [2]”. Reference [2] is not a suitable source for this information.
-What is the purpose of Table 1? The information provided is incomplete to be considered the major differences between the two cell types.
2- One of the aims of the manuscript is to differentiate EOS and Ne cells using resonance Raman spectroscopy and taking the Raman bands of EPO and EMO as markers of the respective cells. Their analysis suggests that the best differentiation of the peroxidases and cells can be done when excitation with 633 nm laser line is used. As mentioned in the introduction part, previous studies have been carried out with this purpose:
On line 73 the authors wrote: “A few reports have shown this approach by illuminating EOS and Ne with laser lines at 406, 514, and 660 nm [8,17,18].”
However, in the manuscript, there is neither critical mention nor comparison with previously published works. For example, in reference 17 (Puppels, G.J.; Garritsen, H.S.; Segers-Nolten, G.M.; de Mul, F.F.; Greve, J. Raman microspectroscopic approach to the study of human granulocytes. Biophys. J. 1991, 60, 1046–1056, doi:10.1016/S0006- 3495(91)82142-7.) Similar approach was applied and they were able to differentiate the EOS and Ne cells by using Raman spectroscopy and excitation with 660 nm. It would be crucial for the manuscript that the authors indicate what is new and what is the improvement of their method to discriminate the cells in comparison with the work done by Puppels et al.
3- On line 338 the authors wrote: “… However, 633 and 785 nm excitations provided significantly different Raman spectra with a specific pattern for each protein and therefore both excitation wavelengths can be employed for unambiguous recognition of EPO and MPO and further for differentiation of granulocytes.”
I agree that EPO and MPO can be differentiated by their spectra in solution. However, I believe, based in the spectra obtained with 633 nm (Figure 3), that it is difficult to differentiate the granulocytes considering only the bands of EPO and MPO. It seems from the figure that the only bands of EPO and MPO that can be used as markers of granulocytes are those at 1564 (EPO) and 1580 (MPO) (see next comment). Other bands of EPO and MPO occur at similar positions and/or overlap with bands corresponding to other biomolecules.
4- There are some weak points about the absence of some bands in the spectra of EPO and MPO as shown in Table 2. For example, let us look at the spectra in Figure 2C (633 nm). In the table there is a band at 1216 cm-1 for EPO but not for EMO. However, in the spectrum of EMO, it can be seen a shoulder at similar position. In the table there is a band at 1240 cm-1 for EMO but not for EPO. However, in the spectra of EPO, there is a shoulder probably at this position. A similar analysis can be done for the band at 1177 cm-1. In the table there is also a band at 1534 cm-1 for EMO but not for EPO. Without doing spectral deconvolution it is not possible to know certainly whether EPO has a corresponding band at this position (see comment 5b). The spectrum of EOS in figure 3B (633 nm) shows a clear shoulder around 1530 cm-1 may be due to EPO.
5-The authors are careless with the correspondence of band positions in the text and in the figures. For example,
On line 262 the authors wrote: “…(Figure 3B). Clearly, the most characteristic bands, which can be used to distinguish the cells, appeared at 1117 (n22), 1215 (n13), and 1564 cm-1 (n19) in the spectrum of EOS and at 1107 (n22), 1241 (n42), 1529 (n38), and 1580 cm-1 (n2) for Ne.”
Several of these numbers are not the same or are absent in the figure.
6- Some points need to be detailed in “Materials and Methods” section:
a) Line 116: After isolation of neutrophils and eosinophils, how did the authors evaluate the cell purity? This is a very important point and should be included in the paper.
b)-On line 142 the authors wrote: “In order to avoid photothermal degradation of peroxidases, the laser power was adjusted individually for each reference sample…”
I suggest the authors to explicitly describe which criteria they have used to choose the laser power to avoid degradation.
c)- it is not mentioned how the band positions were obtained. I guess it was made by deconvolution as the Raman spectra contain several overlapping bands and shoulders.
d) It is missing an Ethics statement about blood donation.
7- In Figure 5, the images provided with fluorescence microscopy are very small and have poor quality. After staining with BODIPY, it is possible to visualize the punctate staining of LBs at high magnification (100x) and I would suggest including the images at this mag. The authors should also provide images taken at the bright field and stained with hematological stains, which clearly differentiate eosinophils from neutrophils. By the images provided, it is impossible to tell which cells are observed. To my view, the results obtained for LBs are overestimated regarding the applicability of the techniques used to study these organelles.
Minor points
-Line 120: the word “microscopy” is misspelled.
-There is a sentence in another language at the beginning of the introduction.
Author Response
Dear Reviewer,
We would like to thank for your comments and suggestions. Please find the answers in the attachment file.

Reviewer 2 Report
The authors reported the unique method to differentiate two types of granulocytes using UV-vis absorption spectroscopy, Raman imaging. In addition, LBs in the cells were assessed by fluorescence and CARS microscopy.
Major
The authors stained eosinophils and neutrophil LBs using BODIPY dye. However, the cells in Fig 5A showed diffuse cytoplasmic staining, indicating the staining itself was not appropriate. The authors should show the specific LB staining by high magnification images, with indicating LBs with arrows for instance. Description for Fig. 5 in result section, especially for Fig.5b and c is insufficient. Is it possible to ascertain how accurate the different LB evaluation methods (i.e. BODYPY and CARS) are?
In line 385, the description starting with “LBs were not formed” needs some reference. If the authors stimulate cells to increase LBs, is it possible to detect the difference using BODYPY and CARS imaging?
2, The blood was “purchased”, but the authors should describe ethical considerations in the using human samples.
3, In the Fig. 2C, the spectrum of the solution for 785 nm excitation was shown in dashed line. What happen if the solution was measured using 532 and 633 nm? Same as Fig. 3, what about the spectrum of the PBS medium?
4, The author concluded the nucleus segmentation was distinguished using Columbus analysis. However, the nucleus segmentation of eosinophils and neutrophils can vary from cell to cell, or from donor to donor. The authors need to show the proof of the accuracy of their calculation.
Minor
1, Describe in detail the company and other details of the reagent. Which company did you use for the eosinophil isolation?
2, What do you mean by “Błąd! Nie można odnaleźć źródła odwołania”? in introduction?
3, Number the line properly in line 43-44.
4, Please spell out for the rR in line 239.
5, Several “specific” bands in result section were not indicated in figure. Please confirm.
6, Double space after “as well as” in line 364.
7, Please spell out for the NIR in line 176.
Author Response

(The authors gave the same response as above.)

Round 2
Reviewer 1 Report
The authors addressed all major points and I feel the paper was improved. However, there are minor points that still need to be addressed.
Introduction:
My concerns are regarding general information about eosinophils. For example, references 2-4 do not support the information that eosinophil nuclei have 10-20 μm in diameter. Where did the authors get such data? Moreover, although refs 2-4 are excellent reviews, there are more recent reviews about eosinophils that should be considered. For example: Weller PF, Spencer LA. Functions of tissue-resident eosinophils. Nat Rev Immunol. 2017;17(12):746-760.
Table 1 has conceptual mistakes. The functions of eosinophils are complex and these leukocytes are currently considered as multifunctional leukocytes. It is not appropriate to consider its primary function as just antiparasitic; eosinophil nuclei are mostly bilobed but can also be polylobed; the eosinophil granule contents are not just cationic proteins but include cytokines, chemokines, and growth factors and the ability of phagocytosis is present in eosinophils. I definitely suggest deleting the table or to provide more substantial information.
References 4 and 5 are the same (listed twice).
Author Response
Dear Reviewer,
We would like to thank You for valuable comments. The answers are included in the attachment.
Please see the attachment.
Best regards,
Aleksandra Dorosz
